# Differential Health Effects on Inflammatory, Immunological and Stress Parameters in Professional Soccer Players and Sedentary Individuals after Consuming a Synbiotic. A Triple-Blinded, Randomized, Placebo-Controlled Pilot Study

**DOI:** 10.3390/nu13041321

**Published:** 2021-04-16

**Authors:** Carmen Daniela Quero, Pedro Manonelles, Marta Fernández, Oriol Abellán-Aynés, Daniel López-Plaza, Luis Andreu-Caravaca, María Dolores Hinchado, Isabel Gálvez, Eduardo Ortega

**Affiliations:** 1International Chair of Sport Medicine, Faculty of Medicine, Campus de los Jerónimos, Catholic University of Murcia, 30107 Murcia, Spain; pmanonelles@ucam.edu (P.M.); mifernandez2@ucam.edu (M.F.); oabellan@ucam.edu (O.A.-A.); dlplaza@ucam.edu (D.L.-P.); landreu@ucam.edu (L.A.-C.); 2Faculty of Sport, Campus de los Jerónimos, Catholic University of Murcia, 30107 Murcia, Spain; 3Grupo de Investigación en Inmunofisiología, Instituto Universitario de Investigación Biosanitaria de Extremadura (INUBE), University of Extremadura, Av. Elvas s/n, 06006 Badajoz, Spain; mhinsan@unex.es (M.D.H.); igalvez@unex.es (I.G.); orincon@unex.es (E.O.)

**Keywords:** anxiety, immunity, inflammation, prebiotic, probiotic, sedentarism, soccer, stress, synbiotic

## Abstract

The main objective of this research was to carry out an experimental study, triple-blind, on the possible immunophysiological effects of a nutritional supplement (synbiotic, Gasteel Plus^®^, Heel España S.A.U.), containing a mixture of probiotic strains, such as *Bifidobacterium lactis* CBP-001010, *Lactobacillus rhamnosus* CNCM I-4036, and *Bifidobacterium longum* ES1, as well as the prebiotic fructooligosaccharides, on both professional athletes and sedentary people. The effects on some inflammatory/immune (IL-1β, IL-10, and immunoglobulin A) and stress (epinephrine, norepinephrine, dopamine, serotonin, corticotropin-releasing hormone (CRH), Adrenocorticotropic hormone (ACTH), and cortisol) biomarkers were evaluated, determined by flow cytometer and ELISA. The effects on metabolic profile and physical activity, as well as on various parameters that could affect physical and mental health, were also evaluated via the use of accelerometry and validated questionnaires. The participants were professional soccer players in the Second Division B of the Spanish League and sedentary students of the same sex and age range. Both study groups were randomly divided into two groups: a control group—administered with placebo, and an experimental group—administered with the synbiotic. Each participant was evaluated at baseline, as well as after the intervention, which lasted one month. Only in the athlete group did the synbiotic intervention clearly improve objective physical activity and sleep quality, as well as perceived general health, stress, and anxiety levels. Furthermore, the synbiotic induced an immunophysiological bioregulatory effect, depending on the basal situation of each experimental group, particularly in the systemic levels of IL-1β (increased significantly only in the sedentary group), CRH (decreased significantly only in the sedentary group), and dopamine (increased significantly only in the athlete group). There were no significant differences between groups in the levels of immunoglobulin A or in the metabolic profile as a result of the intervention. It is concluded that synbiotic nutritional supplements can improve anxiety, stress, and sleep quality, particularly in sportspeople, which appears to be linked to an improved immuno-neuroendocrine response in which IL-1β, CRH, and dopamine are clearly involved.

## 1. Introduction

Physical inactivity and a sedentary lifestyle, as well as an inadequate diet, are factors closely related to the onset of various diseases such as obesity, diabetes, and colon and breast cancer, as well as certain diseases of the immune system, among others. Furthermore, vigorous and continued exercise could also be a mitigating factor in reducing immunocompetence in people [1].

The use of new nutritional strategies such as the consumption of probiotics, prebiotics, and their combination, synbiotics, are postulated as instruments that could generate a multitude of beneficial responses in human health [2,3,4,5,6]. The main mechanism of action of synbiotics is to use this symbiosis (probiotics–prebiotics) to induce an increase in the survival of the host, as well as the implantation of live microbial dietary supplements in the digestive tract, which selectively stimulate the growth and/or activation of the metabolism of one or a limited number of bacteria that promote health [7].

It has been suggested that the intestinal microbiota can modulate the interaction of the hypothalamic–pituitary–adrenal axis, as well as the excitatory and inhibitory activity of some neurotransmitters (serotonin, gamma-aminobutyric acid, and dopamine) and substances similar to neurotransmitters, as a response to some inflammatory cytokines, especially in response to physical and emotional stress [8,9]. Exercise too modulates the interactions between the immune and stress responses mediated by cytokines and neuroimmunomodulators [10,11,12,13]. Furthermore, another exercise-induced immune system alteration, that is to say acute and chronic changes associated with stress in sportspeople, is the secretion of immunoglobulin A (SIgA) in saliva [14]. Thus, it is plausible to speculate that competitive sport and the inherent stress in it could even modify the effects of synbiotics on immuno-neuroendocrine interactions. In this context, according to Ortega (2016) [12], “anti-inflammatory” (i.e., reduction of inflammatory mediators, such as inflammatory cytokines and cell-mediated innate responses) and “anti-stress” (i.e., reduction of stress mediators, such as stress hormones and neurohormones) responses seem to occur after an exercise session in individuals with inflammatory deregulations, paradoxically having the opposite effect in healthy people. This is synthesized in the term “bioregulatory effect of exercise”, defined as an effect which reduces or prevents any excessive effect of inflammatory mediators and stimulates innate defenses against pathogens [12].

The potential exercise-induced “pro-inflammatory“ responses (i.e., stimulation of inflammatory and innate responses mediated by inflammatory cytokines and innate cells) induce a major protection against infections in healthy people. On the other hand, they could exacerbate some immunophysiological and clinical conditions in people suffering from inflammatory or stress diseases. In addition, “anti-inflammatory” effects induced by certain exercises could compromise the effectiveness of the immune system against pathogens. These can occur if the exercise-induced innate/inflammatory responses are not well “bioregulated” [12,15]. Moreover, according to the literature, some reviews [16,17] argue that the consumption of probiotics, prebiotics, and synbiotics could be effective in improving the performance of athletes by maintaining gastrointestinal and immune function, thus reducing the susceptibility to illness. However, Costa et al. (2017) [18] believe that sport itself could modify the intestinal immune response and gastrointestinal functions, thus modifying the microbiota composition.

Taking all of this into account, and also bearing in mind the interplay in the “gut–brain axis”, the main objective of the present investigation was to identify the effect of a synbiotic containing a mixture of probiotic strains (*Bifidobacterium lactis, Bifidobacterium longum*, and *Lactobacillus rhamnosus*) and the prebiotic fructooligosaccharides in athletes and sedentary people, and their potential varying responses. The study aimed to evaluate the effects on different immunophysiological parameters, such as inflammatory/immune and stress mediators, as well as on metabolic profile, physical activity/sedentary levels, and different aspects of perceived general physical and mental health. To the best of our knowledge, this is the first study evaluating in the same individuals the effect of a synbiotic on the immune and stress response, particularly comparing the response between sedentary persons and athletes. In addition, the paucity of research which analyzes the possible mechanisms of action attributed to the various health benefits that the consumption of probiotics, prebiotics, and synbiotics seem to induce should be highlighted here. We observe that most of the literature studies available demonstrate more conceptual conclusions than directly experimental results. In our opinion, this reinforces the novelty of the present investigation.

## 2. Materials and Methods

### 2.1. The Synbiotic

The synbiotic Gasteel Plus^®^ (Heel España S.A.U laboratories) is a nutritional supplement containing a mixture of probiotic strains: *Bifidobacterium lactis* CBP-001010, *Lactobacillus rhamnosus* CNCM I-4036, *Bifidobacterium longum* ES1, and fructooligosaccharides (200 mg) as a prebiotic. Each stick of Gasteel Plus^®^ (300 mg) included lyophilized bacteria powder, equivalent to ≥1 × 10^9^ colony-forming unit (CFU) and also containing 1.5 mg of zinc, 8.25 µg of selenium, 0.75 µg of vitamin, and maltodextrin as an excipient. Placebo sticks were filled with 300 mg excipient of maltodextrin. The subjects were required to take the sticks once per day during the supplementation period, preferably in the morning, and dissolved in water.

### 2.2. Subjects

The final analyzed sample consisted of 27 male participants, 13 of which were professional soccer players in the Second Division B level of the Spanish National League, as well as 14 sedentary students with low levels of physical activity (≤150 min/week). During the protocol, “experimental death” occurred due to an injury in one of the athlete participants, and two participants were also excluded because they did not comply with the inclusion criteria. Figure 1 shows the flow chart of participants’ eligibility in the study, where both groups were randomly subdivided into two: a group administered with the synbiotic, and a control group which received the placebo. The choice of the sports discipline of soccer was due to the scarcity, or almost non-existence, of studies investigating the effect of a synbiotic between sedentary individuals and soccer players, with the majority of the studies being on runners. Table 1 shows the main characteristics of the participants.

### 2.3. Experimental Design

This investigation was a triple-blinded, randomized, placebo-controlled pilot study designed to identify the possible differing effects of the synbiotic Gasteel Plus^®^ supplementation between sedentary individuals and soccer players. Subjects were asked to maintain, two weeks before and during the study, their regular lifestyle and the participants were prohibited from consuming probiotics, prebiotics, or fermented products (yogurt or other foods) and any medications that could interfere with the study protocol. Presenting injury or illness would result in exclusion from the study. All participants were also asked to provide written informed consent before participating in the study, which had been previously approved by the ethics committee of the Catholic University of Murcia (Spain) following current legislation (CE031810). This study was registered in ClinicalTrials.gov (identifier: NCT04776772: available from website).

On two separate days, the “baseline tests” and “final tests” were conducted. All participants performed a series of tests, before which they had to fast. The order and schedule (8 a.m.) of the tests was the same for the “final test” and the same materials and procedures were used. A period of 30 consecutive days elapsed between the baseline and final tests, during which the participants had to ingest their supplement (synbiotic or placebo). Accelerometers were distributed one week prior to the baseline test and the week after the final test. Blood and saliva sampling were taken early in the morning and questionnaires were filled out on the two testing days. The treatment was carried out during the last fortnight of May 2019 and the first fortnight of June of the same year, coinciding with situations of possible physical and mental stress in both participant groups, being at the end of an examination period, as well as the end of the soccer season.

### 2.4. Objective Determination of Levels of Physical Activity, Sedentary Lifestyle, and Sleep Quality: Accelerometry

The accelerometer used was the Actigraph wGT3X-BT, which is a small and light triaxial accelerometer (4.6 × 3.3 × 1.5 cm, 19 g) with a response frequency of 30 to 100 hertz. This device was used to measure different objective parameters such as physical activity and its intensity, energy expenditure, metabolic equivalents rhythms (MET), weekly steps, sedentary bouts, and sleep latency and efficiency. Participants wore the accelerometer held with an elastic band on the non-dominant wrist for seven consecutive days and without interruption, except for those times of the day in which the correct operation of the device could be compromised (showers or any activity related to water). Subsequently, the files generated by the accelerometer were analyzed through a specific software called Actilife 6 (ActiGraph, LLC, Pensacola, FL, USA).

### 2.5. Determination of Perceived Levels of General Health, Stress, Anxiety, Fatigue, Depression, and Sleep Quality: Questionnaires

Participants had to fill out a series of validated questionnaires to identify possible subjective health and mental states.

The SF-36 Questionnaire [19] is an instrument that provides results about the health status of a general population covering eight scales: physical function, physical role, body pain, general health, vitality, social role, emotional role, and mental health. The scales are ordered so that the higher the score, the better the health status (0 to 100).

The Sleep Quality in Healthy Lifestyle and Personal Control Questionnaire (HLPCQ) [20] is a questionnaire whose main objective is to detect and quantify lifestyle patterns that reflect health empowerment, as evidenced by the levels of stress and of the internal locus of control. It also includes a section with various questions to measure sleep quality where higher scores indicate better sleep quality.

The State-Trait Anxiety Inventory (STAI) is a questionnaire that analyzes the degree of anxiety that each participant shows. It is divided into two parts: trait-anxiety (what they usually or generally felt) and state-anxiety (their expressed emotions at a specific moment), in which higher scores indicate a higher state of anxiety [21].

The Perceived Stress Scale (PSS) is a questionnaire that allows the frequency at which individuals experience certain stressful feelings to be assessed, as well as their thoughts in the previous month [22].

The Brief Fatigue Inventory (BFI) is a screening tool designed to assess the severity and impact of fatigue on daily functioning [23]. The higher the score, the higher the degree of fatigue.

Beck’s Depression Inventory (BDI) was used to find out how the participants had felt during the final week, including the day of the test, with which it was possible to determine whether or not they presented signs of depression. Higher scores indicated higher signs of depression [24].

### 2.6. Blood and Saliva Sampling

Blood samples were collected from the subjects at 8 a.m. and were deposited into collection tubes containing the anticoagulant EDTA and coagulating agents to isolate plasma and serum, respectively. The plasma and serum were centrifuged, respectively, at 1600× *g* and 1800× *g* for 10 min. Serum and plasma samples were coded and refrigerated gradually at −20 °C as they were obtained. Finally, samples were stored at −80 °C until further analysis.

Saliva samples were obtained using a non-invasive method (collection methods— SalivaBio Oral Swab, Salimetrics). Participants were asked not to ingest any type of food or drink with sugars, alcohol, and/or caffeine, as well as tobacco, at least 12 h prior to the tests. Volunteers were asked to open the packaging and remove the sterile swab for proper placement in the mouth under the tongue and were recommended to hold it for at least 2 min, to ensure against fluctuations in the volume of the sample. Immediately after, samples were refrigerated at −20 °C and finally stored at −80 °C until further analysis.

### 2.7. Determination of Metabolic Profile

The determinations for obtaining the lipid and glycemic profile were carried out through standard techniques with the automatic analyzer of clinical chemistry BA 400 (BioSystems) in the SYNLAB laboratories (Diagnosticos Globales S.A.U., Badajoz, Spain).

### 2.8. Determination of Immuno-Neuroendocrine Parameters

For the determination of the pro-inflammatory and anti-inflammatory cytokines studied (IL-1β and IL-10), an instrument based on flow cytometry was used: the LuminexTM 200 System instrument (Luminex Corporation, Austin, TX, USA) using the ProcartaPlex TM Multiplex Immunoassay. Catecholamines, such as dopamine, epinephrine, and norepinephrine, as well as stress hormones, such as serotonin, cortisol, corticotropin-releasing hormone (CRH), and the adrenocorticotropic hormone (ACTH), were analyzed by competitive inhibition enzyme immunoassays (ELISA), using, respectively: Dopamine Research Immunoassay, General Epinephrine (EPI) RD-EPI-Ge-96T and General Noradrenaline (NE) RD-NE-Ge-96T, General 5-Hydroxytryptamine (5-HT) RD-5-HT-Ge, the DetectX Cortisol Immunoassay Kit, human corticotropin-releasing hormone (CRH) RD-CRH-Hu (Kelowna, BC, V1W 4V3, Canada), and the human ACTH (adrenocorticotropic hormone) ELISA Kit (Elabscience, Houston, TX, USA). To determine immunoglobulin A in saliva, samples were analyzed by an indirect enzyme immunoassay kit through the Salivary Secretory IgA Kit (Salimetrics LLC, Carlsbad, CA, USA). The procedures followed the instructions of the manufacturers, and the findings were measured using an ELISA auto analyzer to quantify color intensity (Sunrise, Tecan, Männendorf, Switzerland).

### 2.9. Statistics

Statistical analysis was performed with IBM statistics SPSS v20.0 software (SPSS Inc., Chicago, IL, USA). To verify the normality of the data, the Shapiro–Wilk test was performed. The repeated two-way analysis of variance (ANOVA) was used, followed by Student’s paired and unpaired t-tests to analyze the intervention effect. The values were expressed as mean ± standard deviation (SD) and the significance level was considered when *p* < 0.05.

## 3. Results

### 3.1. Effects of the Synbiotic on Physical Activity Levels, Sedentary Lifestyle, and Sleep Quality Objectively Determined by Accelerometry

Results observed in Table 2 show that the sedentary group administered with placebo obtained some significant differences with respect to baseline values (*p* < 0.05). A decrease in calories, metabolic rate, and intensity level of physical activity are reflected. The synbiotic seems to prevent this situation by avoiding the mentioned decreases and even increasing the consumption of kilocalories (Kcal) in sedentary individuals, although without significant differences. Results from the athlete group show no significant changes (*p* > 0.05) in those who consumed the placebo, while those who followed the protocol with the intake of the synbiotic had significant improvements in sleep efficiency and latency, as well as increases in the consumption of Kcal and METS.

### 3.2. Effects of the Synbiotic on Perceived Levels of General Health, Stress, Anxiety, Fatigue, Depression, and Sleep Quality

Data in Table 3 show that the baseline values are quite similar in the sedentary and athlete groups. Only in the athletes did the synbiotic intervention significantly improve (*p* < 0.05) the perceived general health as determined by the SF-36 questionnaire, but no differences were found in perceived sleep quality, state anxiety, or fatigue.

No significant differences were found in the perceived stress (Figure 2A), trait anxiety (Figure 3A), or depression (Figure 4A) between sedentary people and athletes at basal status (before intervention). Figure 2B and Figure 3B show a decrease in the levels of perceived stress (*p* < 0.01) and anxiety (*p* < 0.05) only in the athlete group administered with the synbiotic. Figure 4B shows a decrease (*p* < 0.05) in perceived depression levels in both groups (sedentary and athlete) after the synbiotic treatment. Subsequently, training only affected the behavior in response to the synbiotic intervention in stress and particularly in anxiety (*p* < 0.05), also when evaluating by the two-way ANOVA test). These effects were not due to a placebo effect (C and D of Figure 2, Figure 3 and Figure 4).

### 3.3. Effects of the Synbiotic on Metabolic Profile

Table 4 shows the results corresponding to blood concentrations of glucose, cholesterol, and triglycerides as measurements of metabolic profile. Firstly, individuals in both the sedentary and athlete groups presented lipid and glycemic levels compatible with normal and healthy ranges. Thus, as expected, the consumption of the synbiotic did not provoke an appreciable or significant effect.

### 3.4. Effects of the Synbiotic on Inflammatory, Immunological, and Stress Parameters

It should be noted that both groups presented healthy baseline levels of the inflammatory and immune parameters analyzed (Figure 5A). No significant differences in the IL-1β concentrations were observed between the sedentary and athlete groups. However, a different behavior (*p* < 0.05) between the two groups was found in response to the synbiotic intervention: while the synbiotic increased (*p* < 0.05) the systemic concentration of IL-1β in the sedentary group, it slightly decreased in the soccer player group (Figure 5B).

No significant variations (except a potential placebo effect in the sedentary group) were found in the IL-10 concentration (Figure 6). There were also no significant differences in the levels of immunoglobulin A between groups, or as a consequence of the intervention (Figure 7).

A lower concentration without significant differences were found in the dopamine concentration of the soccer players group (Figure 8A). However, training affected the response to the synbiotic intervention (*p* < 0.05), since it induced a significant increase (*p* < 0.05) in the dopamine concentration only in the athletes (Figure 8B). This effect cannot be attributable to a potential placebo effect of the intervention (Figure 8D). However, the decrease (*p* < 0.05) in epinephrine levels in the sedentary group administered with the synbiotic compared to their basal levels could potentially be attributed to a placebo effect of the intervention (Figure 9B,C). In addition, as shown in Table 5, there were also no significant changes in norepinephrine. Basal concentration of serotonin was, however, higher (*p* < 0.05) in the athlete group than in the sedentary group, but statistical differences with the synbiotic intervention were not observed (Table 5).

Finally, the results corresponding to CRH are shown in Figure 10. Athletes presented lower systemic concentration of CRH than sedentary volunteers (Figure 10A). In addition, the behavior of CRH secretion in response to the synbiotic intervention was also different (*p* < 0.05) between the athlete group and the sedentary group, decreasing significantly (*p* < 0.05) in the sedentary group with respect to their baseline levels (as also found with the placebo) and increasing slightly in the athlete group (Figure 8B). The latter cannot be attributable to a placebo effect of the intervention (Figure 8D). There were no significant differences in levels of cortisol and ACTH between groups or as a consequence of the intervention (Table 5).

## 4. Discussion

This investigation is presented as the first study that analyzes the possible immuno-neuroendocrine, inflammatory, and health effects in soccer players and sedentary individuals after the consumption of a synbiotic (Gasteel Plus^®^).

Nowadays, there are various, generally suggestion-based studies which have reported that quality of life improves with the consumption of probiotics, prebiotics, and synbiotics [25,26,27,28]. These studies also show that these supplements could modify some physiological functions in humans, such as appetite, sleep, mood, and circadian rhythms, all through metabolites produced by the fermentation of microbes in the intestine [29]. Additionally, some studies indicate that poor sleep quality is associated with poorer mental health and psychological stress [30,31]. The results of the present investigation show objective (measured through accelerometry), beneficial effects in the sleep quality of the group of athletes after the administration of the synbiotic, and said effect is corroborated by the subjective perception results obtained through questionnaires.

It is also well accepted that probiotics and prebiotics can contribute to improve mental health. Messaoudi et al. (2011) [32] reported significant differences in the perceived anxiety between an experimental group administered with probiotics (*Lactobacillus helveticus* R0052 and *Bifidobacterium longum* R0175) compared with the control group, as determined by subjective and perception indexes and measurement scales (HADS, Hospital Anxiety and Depression Scale). Likewise, in another study, significant reductions in depression were demonstrated (determined by the measurement scales LEIDS-r, Leiden Index of Depression Sensitivity) after the use of certain probiotic strains (*Bifidobacterium bifidum* W23, *Bifidobacterium lactis* W52, *Lactobacillus acidophilus* W37, *Lactobacillus brevis* W63, *Lactobacillus casei* W56, *Lactobacillus salivarius* W24, and *Lactobacillus lactis*) compared with the control group [33]. These studies are in agreement, at least partially, with the results in the present investigation which introduces a synbiotic containing a mixture of probiotic strains, including *Bifidobacterium lactis* CBP-001010, *Lactobacillus rhamnosus* CNCM I-4036, *Bifidobacterium longum* ES1, and prebiotic fructooligosaccharides. Similarly, Bravo et al. (2011) [34] observed a significant decrease in depression levels in mice after consuming a probiotic (*Lactobacillus rhamnosus* JB-1) compared with their basal levels. It is also important to note that in our investigation, the effect of the synbiotic intervention induced a different anxiety and stress improved response depending on the level of physical activity, thus suggesting a role for regular physical activity or training interaction in these effects, and confirming the close relationship between exercise, exercise-induced stress, and diet in the context of neuroendocrine interactions [9]. In fact, the use of the synbiotic seems to function bidirectionally, particularly in athletes, in increasing the level of daily physical activity as determined, by accelerometry, through steps counts and estimated METS and kilocalories consumption.

Several investigations, most of them original articles evaluating the use of probiotics, have studied the possible interaction of these supplements in the inflammatory/immune system [2,35,36,37,38], as well as others, although to a lesser extent, with synbiotics [6,39,40]. Results on pro-inflammatory cytokine showed a different behavior between sedentary people and athletes, being pro-inflammatory (increased IL-1ß concentration) only in sedentary people after the synbiotic intervention. Likewise, the synbiotic-induced decrease in the anti-inflammatory cytokine IL-10, observed only in the sedentary group, could also contribute to the pro-inflammatory effect in these individuals, even though it was also observed in the placebo group. These results clearly indicate that the immunobioregulatory effects of non-pharmacological interventions (in this case, also with synbiotic consumption) can be different in sportspeople than in sedentary ones, probably due to the different basal “set point” of the inflammatory cytokines and stress hormones [12,41].

Another immune variable that has gained special attention from researchers is immunoglobulin A (IgA). This biomarker in saliva (sIgA) has been constantly associated with the incidence of infections, where low concentrations or substantial transitory falls are related to an increase in diseases of the upper respiratory tract [42]. Gleeson et al. (2011) [43] concluded that regular intake of a specific probiotic, *Lactobacillus casei*, appears to be beneficial in reducing the frequency of symptoms in the upper respiratory tract, and that this is owing to a maintenance of IgA levels in saliva, concurring with the results of other studies [2,42]. The aforementioned studies also suggest an important finding: the increase in mucosal immunity due to the administration of probiotics would serve to protect against infection due to pathogens that penetrate the mucosa. However, in agreement with our results, Cox et al. (2010) [4] concluded that there were no significant changes in immunoglobulin A in saliva between the placebo group and the experimental group that consumed the probiotic strain *Lactobacillus fermentum*. Nevertheless, it is important also to highlight that the synbiotic did not induce worsening levels of IgA in this research.

As already discussed, and referenced in the present investigation with the synbiotic, there is also evidence that the administration of certain probiotics has beneficial effects on mood and on certain psychological problems such as anxiety, stress, fatigue, and depression [32]. All this could also be closely related to the role that these have in the regulation of the intestinal microbiota, as well as their effect on the hypothalamic–hypophysis–adrenal axis (HHA axis) and on pathways of the nervous system [44]. Intestinal microbiota modulates a series of excitatory and inhibitory neurotransmitters (serotonin, dopamine, and gamma-aminobutyric acid or GABA), as well as other neuromodulators, especially in situations of possible physical and emotional stress [8,45]. As such, diet is considered a key part of the regulatory mechanism of this “gut–brain” communication axis, with probiotics, prebiotics, and synbiotics assuming an important role. The results obtained in this study regarding fatigue in athletes (although without statistically significant differences) could be related to the lower levels of dopamine together with the elevated levels of serotonin, with the synbiotic, versus the placebo, being able to counteract these effects here [46]. In fact, habitual exercise or training modified the synbiotic-induced response in the systemic release of dopamine (athletes versus sedentary people). In addition, the synbiotic-induced decrease in epinephrine levels in the sedentary group could be related to the increase of the concentration of the cytokine IL-1β in this same group, since, in healthy individuals, a decrease in catecholamine concentration stimulates the release of IL-1 by macrophages, as well as inflammatory cytokines through Th1 lymphocytes [12,47].

As found in dopamine concentrations, training also affected the synbiotic-induced response in CRH systemic concentration, which was also lower in the athletes than in sedentary volunteers; thus indicating potentially lower levels of stress in the sportspeople. As such, the significant decrease in systemic CRH levels induced by the synbiotic intervention only in the sedentary group could indicate a reduction of stress induced by this nutrition supplement. In any case, it is interesting to observe how this apparent “bioregulatory behavior” of the synbiotic, between sedentary individuals and athletes with lower baseline levels of CRH, is similar to that observed in the behavior of IL-1β, making it plausible to think that the synbiotic-induced variations in the Sympathetic nervous system (SNS ) (previously discussed with epinephrine), and in the HHA axis, now through variations in CRH, are involved in the bioregulatory effects of the inflammatory response [48]. Nevertheless, changes in CRH concentrations did not induce significant physiological variations in ACTH and cortisol; and since a decrease in CRH was also observed in the “sedentary placebo” group, these results open new windows for future investigations.

There is a lack of methodological specificity in most of the previous literature studies regarding inflammatory and immune markers. The unification of these approaches is therefore necessary to avoid partial interpretations of the results, as well as the evaluation of the physiological and clinical relevance of in vitro and ex vivo effects of probiotics, prebiotics, and synbiotics as a nutritional tool for athletes in particular. It is clear, however, that immuno-neuroendocrine interactions affecting the immune response, mental/physical health, and metabolic regulation contribute to the effects of these supplements. As such, the same effects in sedentary people or sportspeople cannot always be expected.

In this context, increasing the number of participants, together with a longer intervention time, would be necessary to obtain more significant responses. The positive diet modification for gut microbiota is presented as a physiological improvement, not only for athletes and their potential improvements in performance, but also for the general population.

## 5. Conclusions

In conclusion, assuming the possible errors that all generalization entails, we can establish that a nutritional supplement containing a mixture of probiotic strains, such as *Bifidobacterium lactis* CBP-001010, *Lactobacillus rhamnosus* CNCM I-4036, *Bifidobacterium longum* ES1, as well as fructooligosaccharides as a prebiotic, induces an “immuno-neuroendocrine bioregulatory effect”, and is therefore dependent on the basal state of the neuroendocrine and inflammatory response of each individual or population group. According to the present investigation, this mainly involves IL-1β, CRH, and dopamine (and to a lesser extent serotonin) which could influence the synbiotic-induced reduction in perceived levels of anxiety and stress, fatigue and depression, as well as the objective improvement in sleep quality. To study the composition of the intestinal microbiota of the athletes versus the sedentary subjects, and its possible variations after synbiotic intervention, a longer intervention would be necessary (duration being a limitation of the present investigation).

## Figures and Tables

**Figure 1 nutrients-13-01321-f001:**
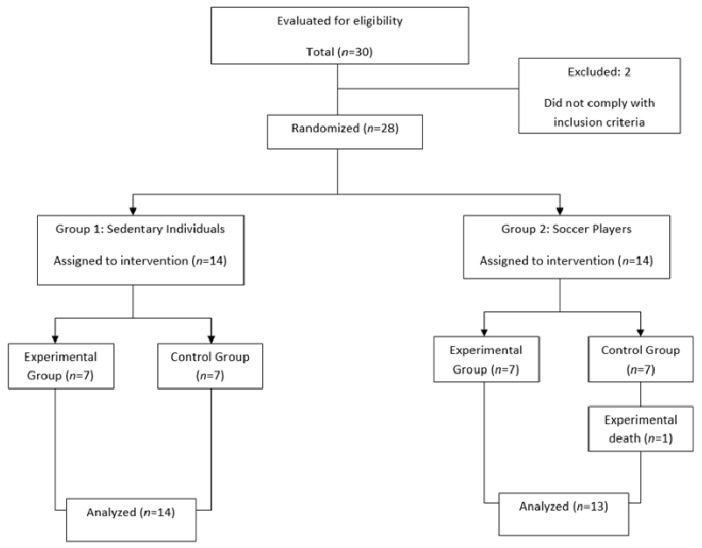
Flow chart of participants’ eligibility.

**Figure 2 nutrients-13-01321-f002:**
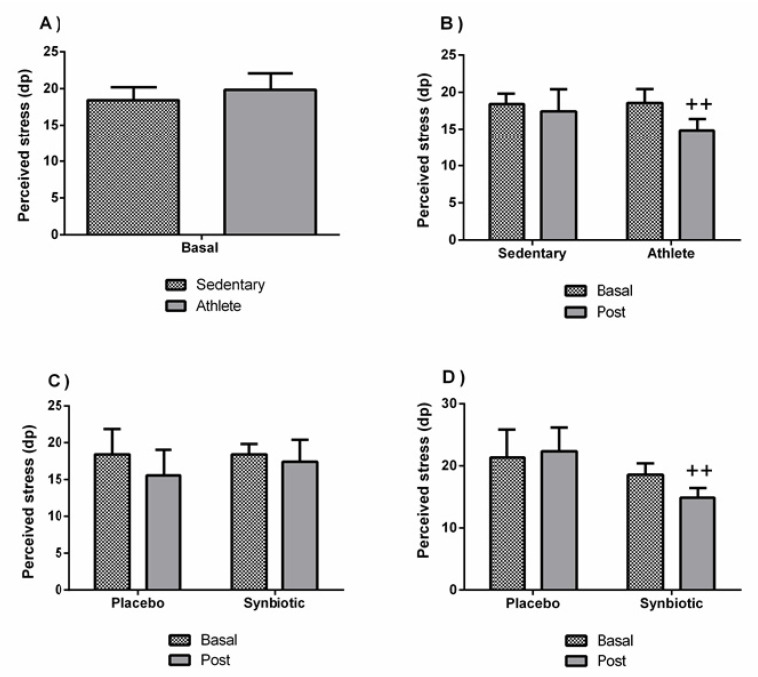
Effect of training and consumption of a synbiotic in this condition on perceived stress levels. (**A**) Perceived stress levels in sedentary men (*n* = 14) and athletes (*n* = 13); (**B**) Influence of training on the effects of the synbiotic on perceived stress levels (*n* = 7 and *n* = 6 in sedentary and athlete groups, respectively); (**C**) Effect of the consumption of the synbiotic on perceived stress levels in sedentary individuals with respect to placebo (*n* = 7) or with synbiotic (*n* = 7); (**D**) Effect of the consumption of the synbiotic on perceived stress levels in athlete individuals with placebo (*n* = 6) or with synbiotic (*n* = 7). The determinations are expressed by the mean ± SD of each of the samples. ++ *p* < 0.01 with respect to the baseline.

**Figure 3 nutrients-13-01321-f003:**
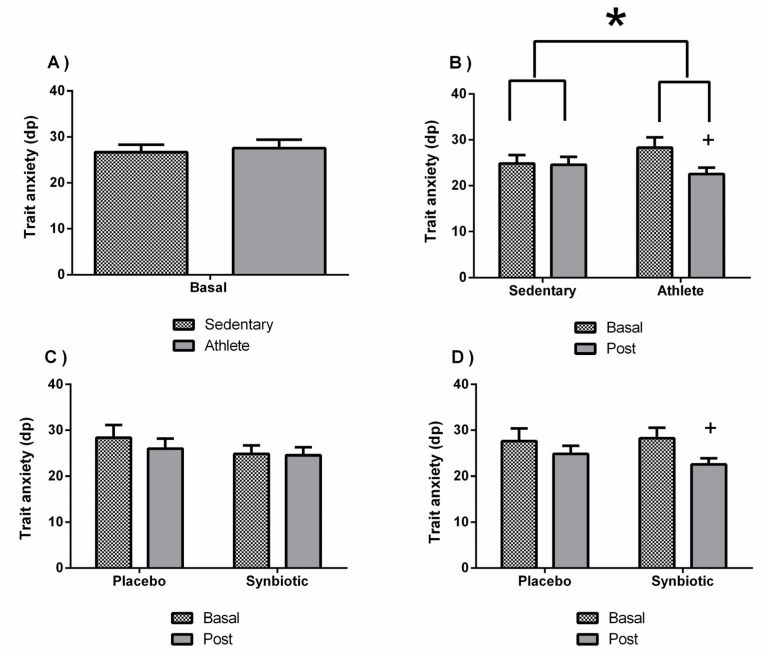
Effect of training and consumption of a synbiotic in this condition on perceived anxiety levels. (**A**) Perceived anxiety levels in sedentary men (*n* = 14) and athletes (*n* = 13); (**B**) Influence of training on the effects of the synbiotic on perceived anxiety levels (*n* = 7 and *n* = 6 in sedentary and athlete groups, respectively); (**C**) Effect of the consumption on trait anxiety levels in sedentary individuals with respect to placebo (*n* = 7) or with synbiotic (*n* = 7); (**D**) Effect of the consumption of the synbiotic on perceived anxiety levels in athlete individuals with placebo (*n* = 6) or with synbiotic (*n* = 7). The determinations are expressed by the mean ± SD of each of the samples. * *p* < 0.05 sedentary group versus athlete group; + *p* < 0.05 with respect to the baseline.

**Figure 4 nutrients-13-01321-f004:**
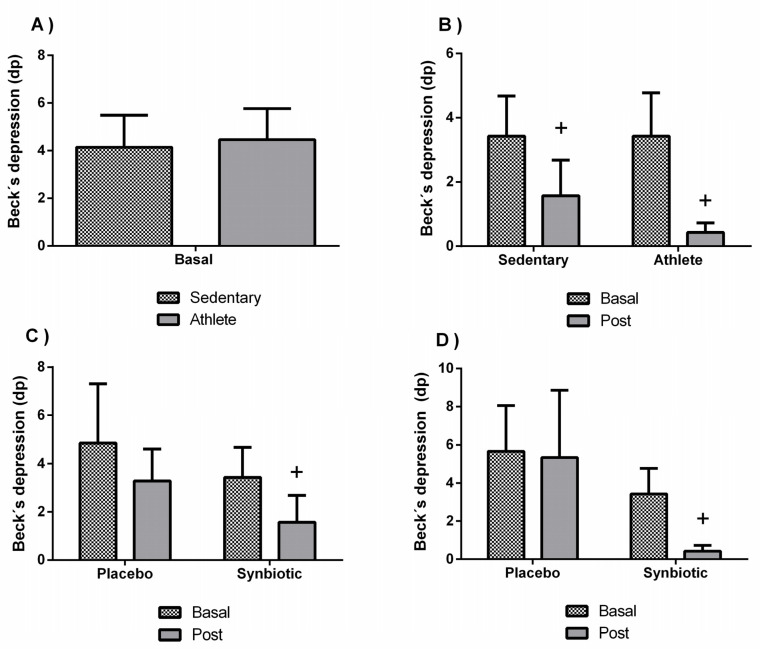
Effect of training and consumption of a synbiotic in this condition on perceived depression levels. (**A**) Perceived depression levels in sedentary men (*n* = 14) and athletes (*n* = 13); (**B**) Influence of training on the effects of the synbiotic on perceived depression levels (*n* = 7 and *n* = 6 in sedentary and athlete groups, respectively); (**C**) Effect of the consumption of the synbiotic on perceived depression levels in sedentary individuals with respect to placebo (*n* = 7) or with synbiotic (*n* = 7); (**D**) Effect of the consumption of the synbiotic on perceived depression levels in athlete individuals with placebo (*n* = 6) or with synbiotic (*n* = 7). The determinations are expressed by the mean ± SD of each of the samples. + *p* < 0.05 with respect to the baseline.

**Figure 5 nutrients-13-01321-f005:**
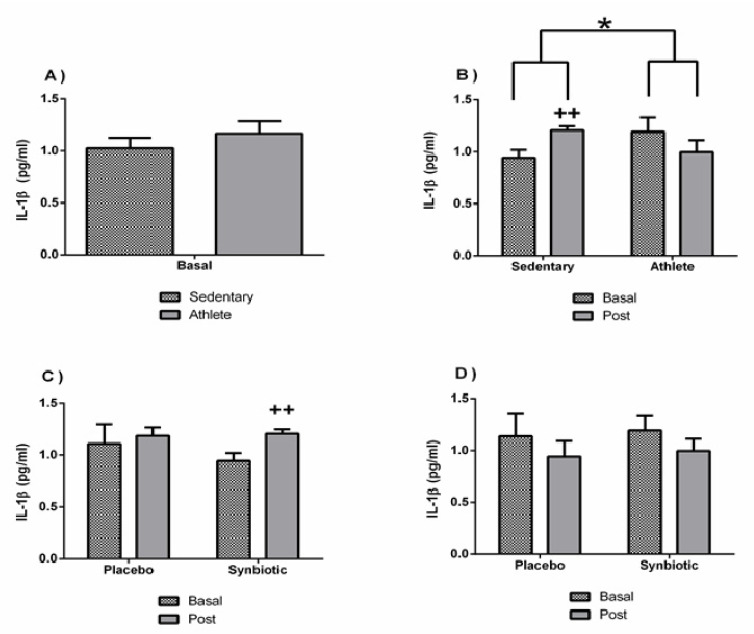
Effect of training and consumption of a synbiotic in this condition on the IL-1β cytokine. (**A**) Basal serum IL-1β concentrations in sedentary men (*n* = 14) and athletes (*n* = 13); (**B**) Influence of training on the effects of the synbiotic on serum IL-1β concentration (*n* = 7 and *n* = 6 in sedentary and athlete groups, respectively); (**C**) Effect of the consumption of the synbiotic on IL-1β in sedentary individuals with respect to placebo (*n* = 7) or with synbiotic (*n* = 7); (**D**) Effect of the consumption of the synbiotic on IL-1β in athlete individuals with placebo (*n* = 6) or with synbiotic (*n* = 7). The determinations are expressed by the mean ± SD of each of the samples. * *p* < 0.05 sedentary group versus athlete group; ++ *p* < 0.01 with respect to the baseline.

**Figure 6 nutrients-13-01321-f006:**
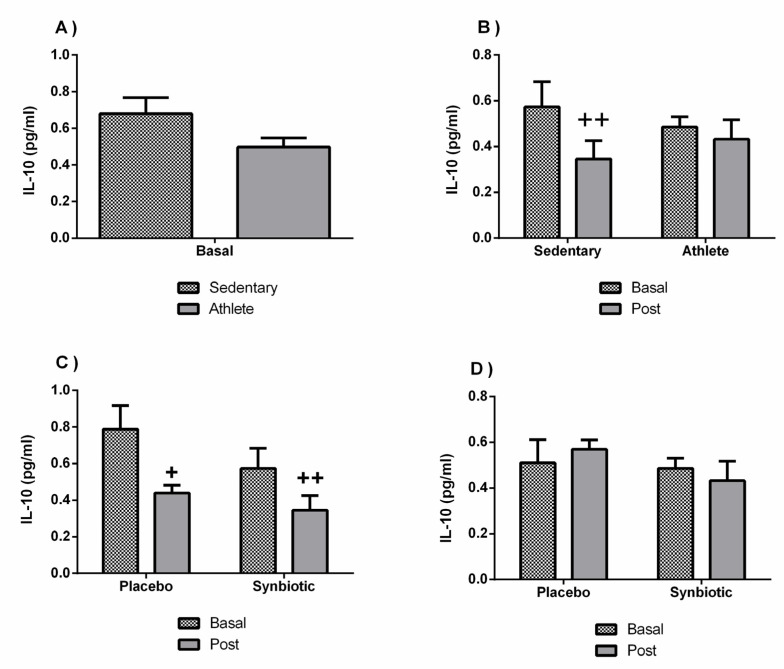
Effect of training and consumption of a synbiotic in this condition on cytokine IL-10. (**A**) Basal serum IL-10 concentrations in sedentary men (*n* = 14) and athletes (*n* = 13); (**B**) Influence of training on the effects of the synbiotic on serum IL-10 concentration (*n* = 7 and *n* = 6 in sedentary and athlete groups, respectively); (**C**) Effect of the consumption of the synbiotic on IL-10 in sedentary individuals with respect to placebo (*n* = 7) or with synbiotic (*n* = 7); (**D**) Effect of the consumption of the synbiotic on IL-10 in athlete individuals with placebo (*n* = 6) or with synbiotic (*n* = 7). The determinations are expressed by the mean ± SD of each of the samples. + *p* < 0.05 and ++ *p* < 0.01 with respect to the baseline.

**Figure 7 nutrients-13-01321-f007:**
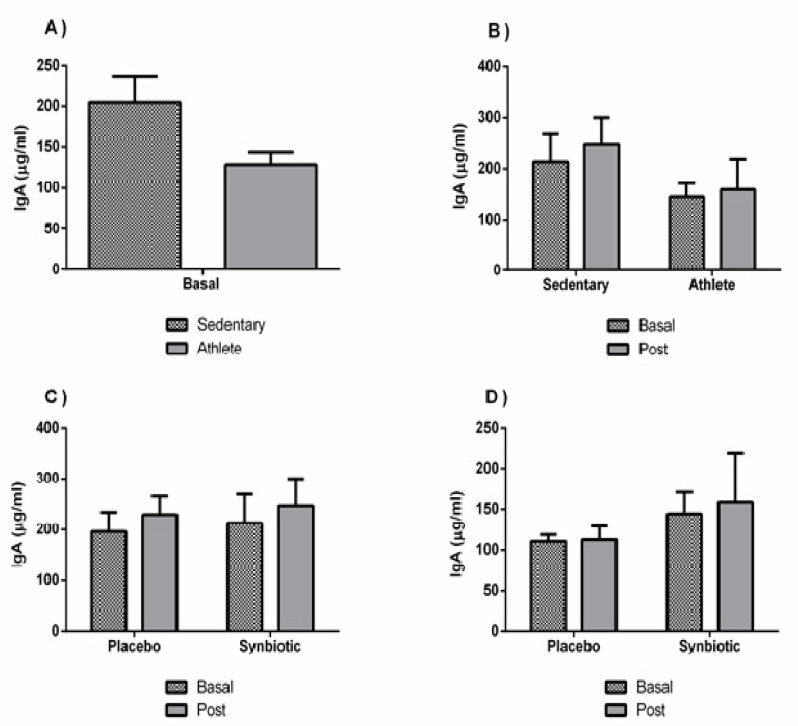
Effect of training and consumption of a synbiotic in this condition on immunoglobulin A (IgA) levels. (**A**) Basal saliva concentrations of IgA in sedentary men (*n* = 14) and athletes (*n* = 13); (**B**) Influence of training on the effects of the synbiotic on the saliva concentration of IgA (*n* = 7 and *n* = 6 in sedentary and athlete groups, respectively); (**C**) Effect of the consumption of the synbiotic on IgA in sedentary individuals with respect to placebo (*n* = 7) or with synbiotic (*n* = 7); (**D**) Effect of the consumption of the synbiotic on IgA in athlete individuals with placebo (*n* = 6) or with synbiotic (*n* = 7). The determinations are expressed by the mean ± SD of each of the samples.

**Figure 8 nutrients-13-01321-f008:**
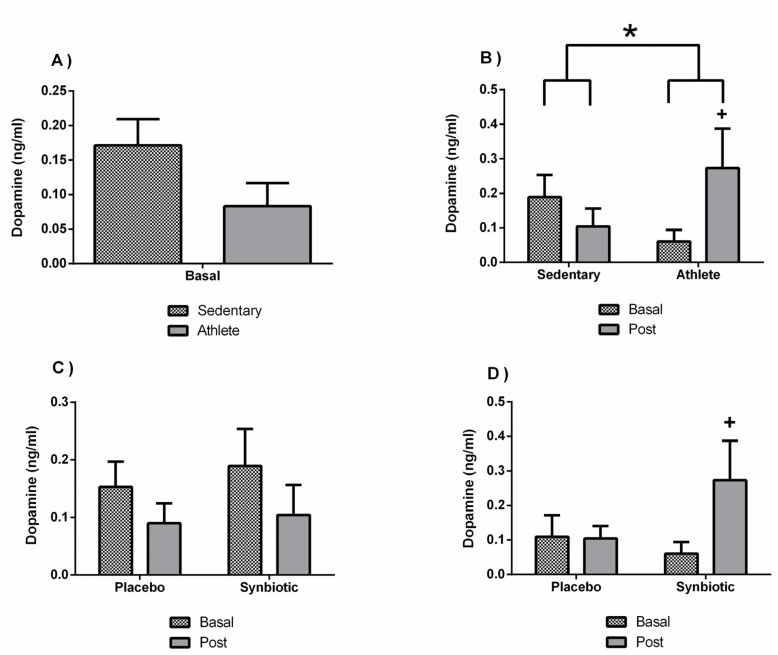
Effect of training and consumption of a synbiotic in this condition on dopamine levels. (**A**) Concentrations of dopamine in sedentary men (*n* = 14) and athletes (*n* = 13); (**B**) Influence of training on the effects of the synbiotic on the dopamine (*n* = 7 and *n* = 6 in sedentary and athlete groups, respectively); (**C**) Effect of the consumption of the synbiotic on dopamine in sedentary individuals with placebo (*n* = 7) or with synbiotic (*n* = 7); (**D**) Effect of the consumption of the synbiotic on dopamine in athlete individuals with respect to placebo (*n* = 6) or with synbiotic (*n* = 7). The determinations are expressed by the mean ± SD of each of the samples. * *p* < 0.05 sedentary group versus athlete group; + *p* < 0.05 with respect to the baseline.

**Figure 9 nutrients-13-01321-f009:**
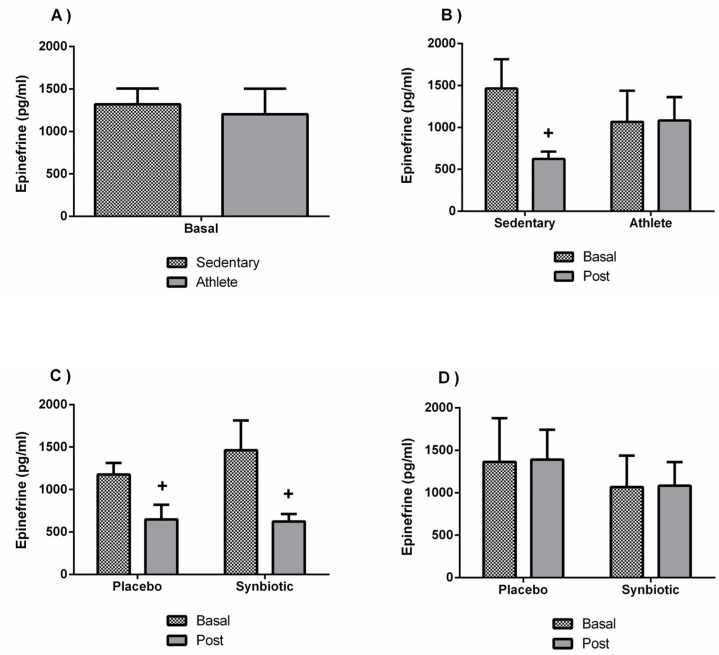
Effect of training and consumption of a synbiotic in this condition on epinephrine levels. (**A**) Concentrations of epinephrine in sedentary men (*n* = 14) and athletes (*n* = 13); (**B**) Influence of training on the effects of the synbiotic on the epinephrine (*n* = 7 and *n* = 6 in sedentary and athlete groups, respectively); (**C**) Effect of the consumption of the synbiotic on epinephrine in sedentary individuals with respect to placebo (*n* = 7) or with synbiotic (*n* = 7); (**D**) Effect of the consumption of the synbiotic on epinephrine in athlete individuals with placebo (*n* = 6) or with synbiotic (*n* = 7). The determinations are expressed by the mean ± SD of each of the samples. + *p* < 0.05 with respect to the baseline.

**Figure 10 nutrients-13-01321-f010:**
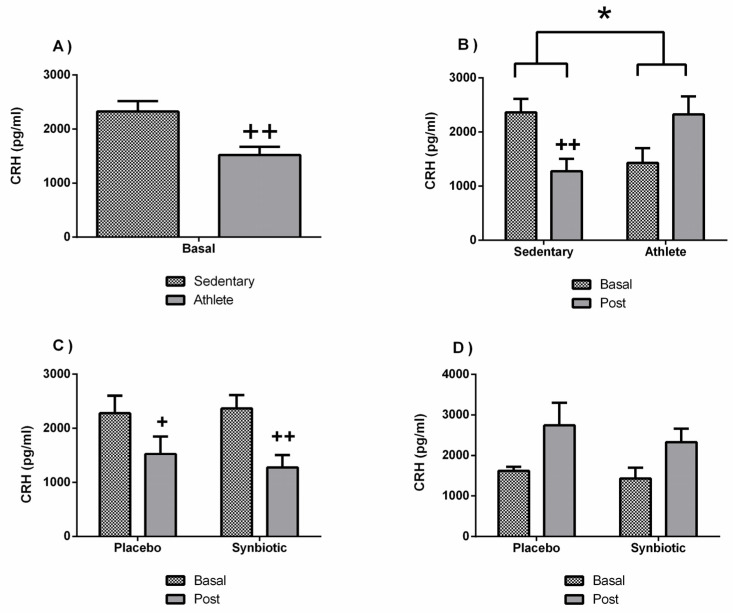
Effect of training and consumption of a synbiotic in this condition on corticotropin-releasing hormone (CRH) levels. (**A**) Concentrations of CRH in sedentary individuals. and athletes (*n* = 13); (**B**) Influence of training on the effects of the synbiotic on the hormone CRH (*n* = 7 and *n* = 6 in sedentary and athlete groups, respectively); (**C**) Effect of the consumption of the synbiotic on CRH in sedentary individuals with respect to placebo (*n* = 7) or with synbiotic (*n* = 7); (**D**) Effect of the consumption of the synbiotic on CRH in athlete individuals with placebo (*n* = 6) or with synbiotic (*n* = 7). The determinations are expressed by the mean ± SD of each of the samples. * *p* < 0.05 sedentary group versus athlete group; + *p* < 0.05 and ++ *p* < 0.01 with respect to the baseline.

**Table 1 nutrients-13-01321-t001:** Descriptive data of the participants.

Sedentary Individuals			Soccer Players	
Variable	Placebo (*n* = 7)	Synbiotic (*n* = 7)	Placebo (*n* = 6)	Synbiotic (*n* = 7)
**Age (years)**	24.31 ± 3.94	23.04 ± 2.09	21.9 ± 2.77	20.66 ± 1.39
**Weight (Kg)**	79.81 ± 8.05	77.47 ± 13.47	73.95 ± 6.42	70.57 ± 6.75
**Height (cm)**	183.97 ± 7.30	176.23 ± 4.49	180.6 ± 8.57	178.23 ± 4.78

The data are represented as mean ± SD.

**Table 2 nutrients-13-01321-t002:** Physical activity levels, sedentary lifestyle, and sleep quality determined by accelerometry.

Sedentary Individuals					Soccer Players			
	Placebo (*n* = 7)		Synbiotic (*n* = 7)		Placebo (*n* = 6)		Synbiotic (*n* = 7)	
Variable	Basal	Post	Basal	Post	Basal	Post	Basal	Post
**Kilocalories (Kcal/week)**	12,185.89 ± 3052.62	10,031.14 ± 209.75 **	9971.39 ± 6062.77	10,311.45 ± 6416.28	8485.81 ± 1572.13	8706.6 ± 2808.90	7856.41 ± 1619.90	8359.98 ± 1590.97 *
**METS (mL O_2_/kg × min)**	1.52 ± 0.13	1.44 ± 0.10 *	1.43 ± 0.21	1.46 ± 0.24	1.4 ± 0.08	1.46 ± 0.16	1.37 ± 0.08	1.49 ± 0.16 *
**MVPA (min)**	1289.85 ± 332.09	1036.57 ± 188.80 *	1081.28 ± 458.38	1081.14 ± 525.82	1091.66 ± 257.39	1103.16 ± 316.51	949.28 ± 231	952.85 ± 227.19
**Steps (total/week)**	81866 ± 11,746.98	66,338.85 ± 7987.85	70,175 ± 17,506.86	67,588 ± 20,406.6	73,096 ± 13,529.34	68,058.66 ± 13,787.82	65,676 ± 11,799.4	689,48.42 ± 12,447.45
**Sedentary bouts (>1 min)**	112.42 ± 17.92	104 ± 16.32	113.71 ± 27.34	114.14 ± 22.32	132.83 ± 22.65	114.16 ± 34.74	127 ± 10.36	120 ± 9.52
**Sleep Latency (min)**	1.12 ± 0.64	1.58 ± 0.83	1.91 ± 1.19	1.55 ± 1.22	0.87 ± 0.49	0.67 ± 0.49	1.38 ± 0.97	0.88 ± 0.74 *
**Sleep efficiency (%)**	87.75 ± 2.87	87.23 ± 3.64	91.44 ± 3.16	91.04 ± 2.18	89.19 ± 3.31	89.6 ± 2.45	87.46 ± 6.09	90.8 ± 3.17 *

* *p* < 0.05 and ** *p* < 0.01 indicate statistically significant difference with respect to the basal values. Data are represented as mean ± SD. METS: metabolic equivalent of task; MVPA: moderate to vigorous physical activity.

**Table 3 nutrients-13-01321-t003:** Perceived levels of general health, state anxiety, fatigue, and sleep quality.

Sedentary Individuals					Soccer Players			
	Placebo (*n* = 7)		Synbiotic (*n* = 7)		Placebo (*n* = 6)		Synbiotic (*n* = 7)	
Variable	Basal	Post	Basal	Post	Basal	Post	Basal	Post
**SF-36**	78.38 ± 13.82	79.2 ± 12.89	77.34 ± 8.62	79.45 ± 8.78	79.36 ± 9.95	80.72 ± 11.39	81.21 ± 9.01	88.5 ± 5.96 **
**Sleep Quality (HLPCQ)**	5.28 ± 2.42	5.14 ± 1.06	6.14 ± 1.95	5.71 ± 2.13	5.16 ± 2.78	5.66 ± 3.14	5.71 ± 2.21	6.42 ± 1.98
**State anxiety (STAI)**	30.14 ± 3.28	30.57 ± 3.9	27 ± 4.54	27.28 ± 3.45	29 ± 5.25	29.83 ± 2.71	30.85 ± 6.86	26.28 ± 6.57
**Brief Fatigue Inventory (BFI)**	2.92 ± 2.12	3.38 ± 2.04	2.98 ± 1.78	1.72 ± 0.63	5.55 ± 3.01	3.43 ± 2.53	3.5 ± 2.2	2.45 ± 1.92

** *p* < 0.01 indicate statistically significant difference with respect to the basal values. Data are represented as mean ± SD.

**Table 4 nutrients-13-01321-t004:** Metabolic profile results.

Sedentary Individuals					Soccer Players			
	Placebo (*n* = 7)		Synbiotic (*n* = 7)		Placebo (*n* = 6)		Synbiotic (*n* = 7)	
Variable	Basal	Post	Basal	Post	Basal	Post	Basal	Post
**Glucose (mg/dL)**	82 ± 6.45	79.71 ± 7.25	87.42 ± 9.6	81.85 ± 9.33	89 ± 6.09	86.33 ± 6.05	88.28 ± 7.88	83.57 ± 6.39
**Total Cholesterol (mg/dL)**	169.28 ± 20.61	169.57 ± 15.95	154.71 ± 31.23	154.14 ± 29.82	143.5 ± 17.09	141 ± 28.93	171.14 ± 22.93	164.85 ± 21.25
**Triglycerides** **(mg/dL)**	96.85 ± 37.72	80.42 ± 33.52	60.14 ± 23.83	60 ± 27.09	40.83 ± 19.34	56.16 ± 31.13	61.85 ± 23.31	78.71 ± 44.35

The data are represented as mean ± SD.

**Table 5 nutrients-13-01321-t005:** Results on immune and stress biomarkers not affected by the synbiotic.

Sedentary Individuals					Soccer Players			
	Placebo (*n* = 7)		Synbiotic (*n* = 7)		Placebo (*n* = 6)		Synbiotic (*n* = 7)	
Variable	Basal	Post	Basal	Post	Basal	Post	Basal	Post
**Cortisol (µg/dL)**	16.53 ± 5.45	15.43 ± 5.19	14.6 ± 5.12	14.39 ± 6.29	16.29 ± 7.29	16.92 ± 4.4	10.65 ± 6.57	14.49 ± 4.74
**ACTH (pg/mL)**	869.07 ± 657.69	512.01 ± 250.23	848.29 ± 481.56	839.96 ± 515.46	733.82 ± 369.02	834.72 ± 389.89	1000.21 ± 802.89 ±	953.68 ± 531.82
**Serotonin (ng/mL)**	34.19 ± 58.38	35.19 ± 72.27	29.17 ± 20.92	15.23 ± 16.32	98.09 ± 123.3 *	79.06 ± 95.94	171.3 ± 211.99 *	81.35 ± 85.81
**Norepinephrine (pg/mL)**	3766.19 ± 429.55	3410.59 ± 573.88	3937.12 ± 348.35	3474.2 ± 340.75	3964.81 ± 355.3	3874.09 ± 416.97	4085.99 ± 171.22	3743.15 ± 391.81

* *p* < 0.05 indicates a statistically significant difference with respect to the sedentary individuals basal levels. The data are represented as mean ± SD. ACTH: adrenocorticotropic hormone.

## Data Availability

The raw data supporting the conclusions of this manuscript will be made available by the authors, without undue reservation, to any qualified researcher.

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
