# Peer review of "Differential Health Effects on Inflammatory, Immunological and Stress Parameters in Professional Soccer Players and Sedentary Individuals after Consuming a Synbiotic. A Triple-Blinded, Randomized, Placebo-Controlled Pilot Study"

_nutrients, 2021, doi:10.3390/nu13041321_

Round 1
Reviewer 1 Report
The research is correctly structured methodologically. However, they have significant shortcomings: the description of the methods lacks reliable information on the calculation of the work performed. Basing the determination of the energy cost on the basis of the accelerometer is scientifically difficult to justify. This is an indirect method that can be used in population studies. The introduction of this method to the assessment of athletes' activity is a misconception. The result of a lack of understanding of sports activity is the statement:
432-435 In fact, the use of the synbiotic to function bidirectionally, particularly in athletes, in increasing the level of daily physical activity as measured by METS and Kilocalories consumption, determined by accelerometry.
The authors do not understand that an athlete's activity is determined by the training program and the intensity of the sports fight. The first component is conditioned by the tasks of the training cycle. The second component largely depends on the activity of the opponent. If the authors write that the probiotic had an influence on the level of daily activity, it would rather be considered whether this is the desired action. If the same tasks are accompanied by increased activity and thus a higher energy cost, the effect is unfavorable. This statement and the interpretation of the research results require a detailed explanation. The work should also be supplemented with a record of physical activity in the list of GPS system registrations that are commonly used in football players training. The level of significance of differences between the analogous values between the control (placebo) and experimental groups was not determined. In a number of indices, the difference between the values at the BASAL level (placebo and experimental groups) is so high. Subsequent inference at the POST level in such cases is unjustified. The experimental data should be supplemented with the level of significance of differences between the analogous values in the placebo and experimental groups. There is no justification in the text for the choice of a sport discipline. This requires justification. There is no justification for the use of the SF 36 questionnaire in 6-person populations, especially in assessing fatigue, stress and quality of life of athletes. This questionnaire cannot be used to evaluate athletes subjected to psychological stresses related to combat sports. There is no validation data for this tool for professional sport. The work presented for review has two components. Component related to physical activity as well as fatigue and stress of the respondents. Biochemical component. The first component requires significant additions in terms of data and statistical analysis. The second component is very well done in terms of methodology. The methods used cannot be called into question.
Author Response
Firstly, we would like to thank you for your time and effort in evaluating the manuscript. Thank for your positive considerations, and we sincerely appreciate all of your helpful, detailed comments and suggestions.
Point 1: The research is correctly structured methodologically. However, they have significant shortcomings: the description of the methods lacks reliable information on the calculation of the work performed. Basing the determination of the energy cost on the basis of the accelerometer is scientifically difficult to justify. This is an indirect method that can be used in population studies. The introduction of this method to the assessment of athletes' activity is a misconception. The result of a lack of understanding of sports activity is the statement:
432-435 In fact, the use of the synbiotic to function bidirectionally, particularly in athletes, in increasing the level of daily physical activity as measured by METS and Kilocalories consumption, determined by accelerometry.
The authors do not understand that an athlete's activity is determined by the training program and the intensity of the sports fight. The first component is conditioned by the tasks of the training cycle. The second component largely depends on the activity of the opponent. If the authors write that the probiotic had an influence on the level of daily activity, it would rather be considered whether this is the desired action. If the same tasks are accompanied by increased activity and thus a higher energy cost, the effect is unfavorable. This statement and the interpretation of the research results require a detailed explanation. The work should also be supplemented with a record of physical activity in the list of GPS system registrations that are commonly used in football players training. The level of significance of differences between the analogous values between the control (placebo) and experimental groups was not determined. In a number of indices, the difference between the values at the BASAL level (placebo and experimental groups) is so high. Subsequent inference at the POST level in such cases is unjustified. The experimental data should be supplemented with the level of significance of differences between the analogous values in the placebo and experimental groups.
Response 1: We understand, of course, that the activity of athletes are subjected to many factors (also including the sport discipline). We agree that accelerometry is probably not the best methodology to evaluate the energy cost and activity in sportspeople during the match or training time. However, our objective and interest was not this, because we hypothesized that a synbiotic intervention (due to the additional modification of other immunophyiological, immune and stress parameters) could also affect differentially the objective daily physical activity and sedentary lifestyle (not during the time of competition) and sleep quality (also linked to activity) between sedentary people and athletes. Then, we need to use the same methodology in the two experimental groups…if not, the scientific experimental design would not be correct.
Nevertheless, and following your right suggestion and criticisms, we have modified the sentence (line 432-435) as follow:
“in increasing the level of daily physical activity as determined, by accelerometry, through steps counts and estimated METS and Kilocalories consumption”. We also feel that was incorrect and now is clarified. Thank you.
Point 2: There is no justification in the text for the choice of a sport discipline. This requires justification.
Response 2: The choice of the sports discipline of soccer was due to the scarcity, or almost non-existence, of studies investigating the effect of a synbiotic between sedentary individuals and soccer players, with the majority of the studies being on runners. It has been included in methods (lines 117-120).
Point 3: There is no justification for the use of the SF 36 questionnaire in 6-person populations, especially in assessing fatigue, stress and quality of life of athletes. This questionnaire cannot be used to evaluate athletes subjected to psychological stresses related to combat sports. There is no validation data for this tool for professional sport.
Response 3: The same justification (as indicated for acelerometry) could be the use of SF-36 for comparing the two experimental groups. It would not be correct to use different methodologies in the same research comparing the effects of an intervention. In any case, the inclusion of the SF-36 questionnaire was carried out to determine, subjectively, the general health perception of each of the participants, being used among the general population and athletes (Antunes et al., 2006; Moreira et al., 2014; Tanabe et al., 2010). Moreover, to analyze fatigue, stress and quality of life other questionnaires were used (the Healthy Lifestyle and Personal Control Questionnaire, the Perceived Stress Scale and the Brief Fatigue Inventory).
Antunes, H. K., Andersen, M. L., Tufik, S., & De Mello, M. T. (2006). Physical stress and physical exercise dependence. Revista Brasileira de Medicina do Esporte, 12(5), 234-238.
Moreira, N. B., Vagetti, G. C., de Oliveira, V., & de Campos, W. (2014). Association between injury and quality of life in athletes: A systematic review, 1980–2013. Apunts. Medicina de l'Esport, 49(184), 123-138.
Tanabe, T., Snyder, A. R., Bay, R. C., & Valovich McLeod, T. C. (2010). Representative values of health-related quality of life among female and male adolescent athletes and the impact of gender. Athletic Training & Sports Health Care, 2(3), 106-112.
You right with the scarce number of volunteers (further studies are proposed with a high number of volunteers and long-time of intervention, as indicated in the manuscript, from this pilot study).
Thank you again for your criticisms and suggestions. We hope we have clearly clarified our methodology design and the justification for using the same methodology for the two experimental group in order to our objectives.

Reviewer 2 Report
It is a very well-written manuscript. The figures and tables are clear. Overall scientific findings are well presented.
I agree with the authors that a longer intervention would be necessary because the duration is being a limiting factor in this study.
Author Response
Point 1: It is a very well-written manuscript. The figures and tables are clear. Overall scientific findings are well presented.
I agree with the authors that a longer intervention would be necessary because the duration is being a limiting factor in this study.
Response 1: We would like to thank you for your time and effort in evaluating the manuscript. Thank you very much for your positive consideration and evaluation of our investigation.

Reviewer 3 Report
Dear Authors,
I appreciate the content of your work, that it is innovative, well conducted and deserve publication.
I would only suggest to clarify the concept of pro-inflammatory and anti-stress, as well as anti-inflammatory.
Sentences from line 62 to 80 of page 2 require revision to clarify the concept of anti or pro inflammatory effect of exercise. I would probably define the terms and stay with them.
Similarly, the paragraph from line 436 to 451 page 17 needs to be clarified.
Author Response
Firstly, we would like to thank you for your time and effort in evaluating the manuscript. Thank for your positive considerations and we sincerely appreciate all of your helpful, detailed comments and suggestions.
Point 1: I would only suggest to clarify the concept of pro-inflammatory and anti-stress, as well as anti-inflammatory.
Sentences from line 62 to 80 of page 2 require revision to clarify the concept of anti or pro inflammatory effect of exercise. I would probably define the terms and stay with them.
Response 1:
As you mentioned on page 2, from line 62 to line 80 of the manuscript, and in page 17, we have included a clarification of the terms ''pro-inflammatory'', ''anti-inflammatory'' and ''anti-stress'', as follows:
“Anti-inflammatory” (i.e. reduction of inflammatory mediators, such as inflammatory cytokines and cell-mediated innate responses); “anti-stress” (i.e. reduction of stress mediators such as stress hormones and neurohormones); “pro-inflammatory” (i.e. stimulation of inflammatory and innate responses mediated by inflammatory cytokines and innate cells).
Point 2: Similarly, the paragraph from line 436 to 451 page 17 needs to be clarified.
Response 2:
Paragraph from line 436 to 451 has also been clarified in the revised version of the manuscript:
We have modified the sentence …“in increasing the level of daily physical activity as measured by METS and Kilocalories consumption, determined by accelerometry” by “ in increasing the level of daily physical activity, as determined by accelerometry, through steps counts and estimated METS and Kilocalories consumption”.
We have also clarified the next paragraph (line 439-451) as:
“Results on pro-inflammatory cytokine showed a different behavior between sedentary people and athletes; being pro-inflammatory (increased IL-1ß concentration) only in sedentary people after the synbiotic intervention. Likewise, the synbiotic-induced decrease in the anti-inflammatory cytokine IL-10, observed only in the sedentary group, could also contribute to the pro-inflammatory effect in these individuals, even though it was also observed in the placebo group. These results clearly indicate that the inmunobioregulatory effects of non-pharmacological interventions (in this case also with synbiotic consumption) can be different in sportspeople than in sedentary ones, probably due to the different basal “set point” of the inflammatory cytokines and stress hormones” [12, 41].

Round 2
Reviewer 1 Report
I appreciate the authors' efforts to supplement the presented text. In my opinion, the amendments and additions should be much larger. The justification for the choice of a group of footballers with the lack of publications on this subject is not right. I propose to get acquainted with the work: https://doi.org/10.1155/2015/783761 Another question arises. Whether professional footballers at the decisive moment of the season could completely give up other supplements and energy? They certainly couldn't. Therefore, these measures should have been included in the list of supplements. Whether they are neutral, they support or neutralize symbiotics. I propose what the description of supplementation and training activities for footballers should look like in the publication: https://doi.org/10.3390/ijerph17228567 There is still a reference to footballers in the title of the work. Therefore, justifying the choice of methods with the criterion of accessibility for students is not relevant. If information was to be obtained about the effect of supplementation on people with low activity, the control group had to be selected from the same population with high activity. In this case, the use of an accelerometer would be justified. It was also possible to use an experiment in which some participants are subjected to systematic physical loads. Can non-athletes be the control group for professional athletes? Yes, but when we are looking for parameters conducive to a sports career. In the discussed case, athletes are a group after many years of selection. Therefore, the observation of the effect of a symbiotic should refer to its interaction with specific states for this group (e.g. after training of a specific energetic nature). The same is true for assessing your stress load. The juxtaposition of trainers and non-trainers is wrong in itself. The reference to participation in the examination session as a sports competition stressor is very original. Authors should provide papers in the field of psychology in which such methodology was used. Re-translation of the use of methods by accessibility to the control group (as in the case of manual workload) cannot be positively assessed.
Author Response
Firstly, we would like to thank you again for your time and effort in evaluating the manuscript. Thank for your enriching comments, and we sincerely appreciate all of your helpful, detailed comments and suggestions.
Point 1: The justification for the choice of a group of footballers with the lack of publications on this subject is not right. I propose to get acquainted with the work: https://doi.org/10.1155/2015/783761 Another question arises. Whether professional footballers at the decisive moment of the season could completely give up other supplements and energy? They certainly couldn't. Therefore, these measures should have been included in the list of supplements. Whether they are neutral, they support or neutralize symbiotics. I propose what the description of supplementation and training activities for footballers should look like in the publication: https://doi.org/10.3390/ijerph17228567.
Response 1:
As you have suggested, and referring to the article by Baralic et al. (2015) in which the following paragraph is included:
“They agreed to avoid the use of vitamin/mineral supplements, antioxidant supplements, nutritional supplements, ergogenic aids, herbs, and medications known for their effect on immune function, 1 month before and during the study”.
In our study, the subjects had to respond, two weeks before the investigation that they were not consuming any type of supplement or medication that could interfere with the consumption of the synbiotic, as well as during the research period. Two subjects were excluded because they were taking antibiotics at the time of the study. In addition, the players had a nutritionist who ensured that their diet was controlled, however, in our investigations no supplements were reported to have been consumed as suggested in the article by Ksiazek et al (2020), being this article posterior to our research, and it would certainly have been very useful for our methodology.
That is why, as you suggest, we have decided to include the following statement in our article (lines 131-134):
“Subjects were asked to maintain, two weeks before and during the study, their regular lifestyle and the participants were prohibited from consuming probiotics, prebiotics or fermented products (yogurt or other foods) and any medications that could interfere with the study protocol”.
Point 2: There is still a reference to footballers in the title of the work. Therefore, justifying the choice of methods with the criterion of accessibility for students is not relevant. If information was to be obtained about the effect of supplementation on people with low activity, the control group had to be selected from the same population with high activity. In this case, the use of an accelerometer would be justified. It was also possible to use an experiment in which some participants are subjected to systematic physical loads. Can non-athletes be the control group for professional athletes? Yes, but when we are looking for parameters conducive to a sports career. In the discussed case, athletes are a group after many years of selection. Therefore, the observation of the effect of a symbiotic should refer to its interaction with specific states for this group (e.g. after training of a specific energetic nature).
Response 2:
In our opinion, the research methodology is being misunderstood, because it is not considering the group of athletes as an experimental group and the group of sedentary people as a control group, but within each group, as indicated in the experimental design, there is a symbiotic group (experimental) or placebo (control), with most of the results being behavioural changes depending on the supplement and the group (athlete or sedentary). We can observe in the literature several investigations that have carried out a protocol similar to ours:
Patlar, S., Yalçin, H., & Boyali, E. (2012). The effect of glycerol supplements on aerobic and anaerobic performance of athletes and sedentary subjects. Journal of Human Kinetics, 34, 69. DOI: 10.2478/v10078-012-0065-x.
Çınar, V., Talaghir, L. G., Akbulut, T., Turgut, M., & Sarıkaya, M. (2017). The effects of the zinc supplementation and weight trainings on the testosterone levels. Человек. Спорт. Медицина, 17(4). DOI: 10.14529/hsm170407.
Cinar, V., Mogulkoc, R., Baltaci, A. K., & Polat, Y. (2008). Adrenocorticotropic hormone and cortisol levels in athletes and sedentary subjects at rest and exhaustion: effects of magnesium supplementation. Biological trace element research, 121(3), 215-220. DOI: 10.1007/s12011-007-8052-0.
Point 3: The same is true for assessing your stress load. The juxtaposition of trainers and non-trainers is wrong in itself. The reference to participation in the examination session as a sports competition stressor is very original. Authors should provide papers in the field of psychology in which such methodology was used.
Response 3:
We would like to emphasize once again that our main objective is not to evaluate sedentary individuals simply as a control group of football players, but to know the effect of the synbiotic in the two experimental groups in “a real life situation” by determining if there are differences between pre and post measurements, as well as between groups, but always taking into account the basal levels of each group, as reflected throughout the research and highlighted in lines 30 and 31 of the manuscript abstract.
Thank you again for your criticisms and suggestions. We hope we have clearly clarified our methodology design and the justification for using the same methodology for the two experimental group in order to our objectives.
